# TP53 Mutation Predicts Worse Survival and Earlier Local Progression in Patients with Hepatocellular Carcinoma Treated with Transarterial Embolization

**DOI:** 10.3390/curroncol32010051

**Published:** 2025-01-18

**Authors:** Ken Zhao, Anita Karimi, Luke Kelly, Elena Petre, Brett Marinelli, Erica S. Alexander, Vlasios S. Sotirchos, Joseph P. Erinjeri, Anne Covey, Constantinos T. Sofocleous, James J. Harding, William Jarnagin, Carlie Sigel, Efsevia Vakiani, Etay Ziv, Hooman Yarmohammadi

**Affiliations:** 1Department of Radiology, Memorial Sloan Kettering Cancer Center, New York, NY 10065, USApetree@mskcc.org (E.P.); marinelb@mskcc.org (B.M.); alexane@mskcc.org (E.S.A.); sotirchv@mskcc.org (V.S.S.); erinjerj@mskcc.org (J.P.E.); coveya@mskcc.org (A.C.); sofoclec@mskcc.org (C.T.S.); zive@mskcc.org (E.Z.); yarmohah@mskcc.org (H.Y.); 2Department of Medicine, Memorial Sloan Kettering Cancer Center, New York, NY 10065, USA; hardinj1@mskcc.org; 3Department of Surgery, Memorial Sloan Kettering Cancer Center, New York, NY 10065, USA; jarnagiw@mskcc.org; 4Department of Pathology, Memorial Sloan Kettering Cancer Center, New York, NY 10065, USA; sigelc@mskcc.org (C.S.); vakianie@mskcc.org (E.V.)

**Keywords:** hepatocellular carcinoma, interventional oncology, transarterial embolization, tumor protein p53

## Abstract

The aim of this study was to evaluate associations between TP53 status and outcomes after transarterial embolization (TAE) for the treatment of patients with hepatocellular carcinoma (HCC). This single-institution study included patients from 1/2014 to 6/2022 who underwent TAE of HCC and genomic analysis of tumoral tissue. The primary outcome was overall survival (OS) with relation to TP53 status, and the secondary outcome was the time to progression. Survival analysis was performed using the Kaplan–Meier method. The time to progression with death or the last patient contact without progression as competing risks were used to obtain a cumulative incidence function, and the association with TP53 status was evaluated using the Gray test. In total, 75 patients (63 men) with a median age of 70.0 (IQR 62.0–76.3) years were included. Of these, 26/75 (34.7%) patients had TP53-mutant HCC. Patients with TP53-mutant HCC had a significantly worse median OS of 15.2 (95% CI, 9.5–29.3) months, versus 31.2 (95% CI, 21.2–52.4) months as the median OS (*p* = 0.023) for TP53 wild-type HCC. Competing risk analysis showed a shorter time to local hepatic progression (at the site of the previously treated tumor) after TAE in patients with TP53-mutant HCC. The cumulative incidences of local progression at 6 and 12 months for TP53-mutant HCC were 65.4% and 84.6%, versus 40.8% and 55.1% for TP53 wild-type HCC (*p* = 0.0072). A TP53 mutation may predict a worse overall survival and a shorter time to local progression in HCC patients treated with TAE.

## 1. Introduction

Hepatocellular carcinoma (HCC) is the third-leading cause of cancer-related deaths worldwide [1]. There are several curative-intent treatments for HCC, including hepatectomy, liver transplantation, percutaneous thermal ablation, and yttrium-90 radiation segmentectomy [2,3,4]. However, many HCC patients are not candidates for curative-intent therapy, in part due to the tumor burden or the severity of their underlying chronic liver disease, and are treated with the intent of disease control [5,6]. Transarterial embolization (TAE) and transarterial chemoembolization (TACE) are locoregional therapies for disease control and appropriate for approximately 20% of HCC patients [6]. These are both HCC treatments with ischemia as the primary mechanism of tumor cell death. The clinical outcomes of TAE and TACE are very similar [7,8,9]. TAE has the advantages of not requiring pharmaceutical preparation, repeatability with preservation of the vascular anatomy, and an approachable learning curve [10,11].

However, TAE of HCC has a variable clinical response that is challenging to predict, and disease recurrence is common [7,12]. There have been efforts to identify a molecular marker or genetic signature predictive of response and prognosis in HCC patients treated with TAE, consistent with growing efforts to personalize treatment in oncology [13,14,15,16]. However, the existing data are scant, partly because histologic confirmation is not always necessary for a diagnosis of HCC, and tissue sampling comes with risks of bleeding and tumor seeding [17].

TP53 is a tumor suppressor gene that is implicated in many malignancies and is frequently mutated in HCC (18–31%) [18,19,20]. There is limited evidence associating a TP53 mutation with a worse prognosis in HCC; however, the data are primarily derived from patients with surgically treated early-stage disease [19,20,21,22,23]. The prognostic value of a TP53 mutation in HCC patients treated with TAE remains yet unknown. The hypothesis is that a TP53 mutation is predictive of worse survival in HCC patients treated with TAE. The purpose of the present study was to evaluate the associations between the TP53 mutational status and clinical outcomes after TAE of HCC.

## 2. Materials and Methods

### 2.1. Patients

This was a single-institution study. Adult patients were either identified through a prospective biospecimen protocol or included from a retrospective database. Both the prospective and retrospective components of this study were compliant with the Health Insurance Portability and Accountability Act.

Prospectively identified patients from January 2014 to June 2022 underwent a biopsy at the time of TAE. Both the prospective collection and use of biospecimens were institutional review board approved, and patients gave their informed consent for the prospective collection. The topic of this study was not the major focus of the aforementioned prospective study.

Retrospectively identified patients underwent TAE from January 2011 to June 2022 and had tumoral tissue collected via either a biopsy or surgical resection. The retrospective collection and analysis of the corresponding data were approved under a separate institutional review board protocol. Consent was waived for this cohort due to the retrospective nature of the study.

All patients had a tissue diagnosis of HCC. Exclusion criteria included a lack of tumoral genetic analysis, non-TAE locoregional therapy, the combination of TAE with another locoregional therapy (e.g., ablation), the combination of TAE with concurrent systemic therapy, a lack of baseline contrast-enhanced imaging, and Barcelona Clinic Liver Cancer (BCLC) stage 0 or D disease.

### 2.2. Transarterial Embolization Procedure

All TAE treatments were part of standard-of-care therapy after a consensus multidisciplinary discussion. The procedures were performed with either conscious sedation or general anesthesia. All TAE procedures were performed by fellowship-trained attending interventional radiologists with 2–30 years of experience. The TAE procedure was performed using a previously described technique [7,24]. Microparticles (Embosphere^®^ Microsphere; Merit Medical, South Jordan, UT, USA) were utilized to embolize the tumoral arterial supply as selectively as possible. The microparticle size was chosen by the performing interventional radiologist, and when deemed safe and appropriate, smaller beads were favored to cause distal arterial occlusion. TAE procedures usually started with smaller size particles (40–120 µm or 100–300 µm), and the particle size was progressively increased as needed until complete stasis was achieved. Complete stasis was defined as contrast opacification of the target vessel without washout for five heartbeats.

### 2.3. Data Collection

The clinical characteristics including age, histopathologic diagnosis, tumoral genetic alterations, presence of synchronous cancer, gender, ethnicity, etiology of liver disease, Eastern Cooperative Oncology Group (ECOG) performance status, history of prior HCC treatment, date of death or loss to follow-up, and laboratory values, including alpha fetoprotein (AFP), bilirubin, and albumin, were obtained from existing maintained databases or a retrospective review of the electronic medical record.

Baseline multi-phasic contrast-enhanced imaging, including CT or MRI, was reviewed to determine the number of tumors, assess for a unilobar or bilobar tumor distribution, assess for macrovascular invasion (tumoral invasion of the portal or hepatic vein), assess for extrahepatic disease, and measure the largest axial dimension of the largest tumor. The largest tumor was designated the index tumor. Each patient’s BCLC stage was determined based on a review of their clinical and imaging characteristics [3].

### 2.4. Genetic Analysis

Formalin-fixed and paraffin-embedded tissue specimens acquired from either percutaneous core needle biopsies or hepatic resections were used. All tissue specimens were examined by pathologists at this study’s institution and analyzed for genetic mutations using IMPACT, a hybridization-capture-based, targeted next-generation sequencing array [25]. For patients with multifocal HCC who underwent a core-needle biopsy, genetic analysis of a single biopsied lesion was performed.

### 2.5. Outcome Assessment

The primary outcome was overall survival (OS) with relation to the TP53 status, as calculated from the time of the first TAE to the time of death or last-known patient contact. The secondary outcome was time to disease progression with relation to the TP53 status, as calculated from the time of the first TAE to the time of progression on the follow-up imaging.

The initial post-TAE cross-sectional imaging was a contrast-enhanced CT or MRI obtained 1 month following the procedure. Subsequently, imaging was typically obtained at 3-month intervals. Two board-certified attending interventional radiologists (K.Z. and H.Y., with 3 and 11 years of experience, respectively) reviewed the cross-sectional imaging for treatment response and disease progression. Discordances were resolved by consensus discussion.

The initial radiographic response of a tumor to TAE was assessed according to the modified Response Evaluation Criteria in Solid Tumors (mRECIST) on the first follow-up imaging study after a complete course of TAE [3]. For patients with a tumor burden necessitating staged TAE treatments, a complete course of TAE was defined as consecutive TAE procedures that treated different arterial territories performed within a 3-month span. For example, bilobar disease treated with right hepatic lobe TAE followed by subsequent left hepatic lobe TAE within 3 months was considered a single complete course of TAE.

Disease progression was defined as “local intrahepatic” for disease that recurred or progressed at the site of a tumor that was previously treated with TAE, “distant intrahepatic” if progression occurred at any site within the liver that did not correspond to the previously treated tumor, “any intrahepatic” for progression at any location within the liver (local or distant), and “extrahepatic” if the disease progressed outside of the liver.

Adverse events related to TAE were assessed based on the Society of Interventional Radiology Adverse Event classification [26]. Events that were grade 2 (moderate) or greater were recorded.

### 2.6. Statistical Analysis

Patients were sub-grouped based on the presence of a tumoral TP53 mutation. Differences in patient characteristics were assessed by Fisher’s exact test for categorical variables. For continuous variables, normality was first assessed using the Shapiro–Wilk test. If normality was confirmed, Student’s *t*-test was used; otherwise, the Mann–Whitney–Wilcoxon test was used. Survival analysis was performed using the Kaplan–Meier method and the log-rank test. Univariate analysis of prognostic factors for improved survival was performed using Cox regression. A univariate *p* lower than 0.05 was used as a cutoff for inclusion within multivariate Cox analysis. The time to progression, with death or the last patient contact without progression as competing risks, was used to obtain a cumulative incidence function. The Gray test was used to evaluate the cumulative incidence function between TP53-mutant and wild-type subgroups [27]. A *p*-value lower than 0.05 was considered statistically significant. Statistical analysis was performed using RStudio (version 6, January 2023, Build 524) [28].

## 3. Results

### 3.1. Patient Characteristics

Prospective accrual identified 219 patients and retrospective review identified 272 patients for potential inclusion in this study. A study flowchart with reasons for patient exclusion is given in Figure 1.

The final cohort included 75 patients (63 men, 12 women) with a median age of 70.0 (IQR 62.0–76.3) years; 40 patients were identified prospectively and 35 retrospectively. Baseline patient and lesion characteristics are summarized in Table 1 and Table 2.

A mutation in TP53 was present in 26/75 (34.7%) tumors. TP53-mutant tumors were significantly more likely to be poorly differentiated (*p* = 0.011) and exhibit baseline macrovascular invasion (*p* = 0.018) than wild-type tumors. Correspondingly, patients with TP53-mutant tumors were significantly more likely to be staged as BCLC C (*p* = 0.007). Other baseline patient and tumor characteristics were not significantly different between the two groups.

### 3.2. Outcomes

The median OS for all patients was 26.1 months (95% CI, 18.3–36.4). Patients with TP53-mutant tumors had a significantly worse median OS of 15.2 months (95% CI, 9.5–29.3), versus 31.2 months (95% CI, 21.2–52.4) (*p* = 0.023) for patients with TP53 wild-type tumors (Figure 2).

Imaging follow-up at 1 month after the first complete session of TAE demonstrated an objective response rate (ORR) of 70/75 (93.3%), inclusive of 36/75 (48%) patients with a complete response (CR) and 34/75 (45.3%) with a partial response (PR) (Table 3). Patients with a CR at 1 month had a significantly longer median OS of 39.2 months (95% CI, 21.2–62.6), versus 16.7 months (95% CI, 9.5–29.3) (*p* = 0.009) for patients without a CR (Figure 3A). Progressive disease (PD) at 1 month was seen in 5/75 patients (6.7%), and patients with PD had a significantly shorter median OS of 3.5 months (95% CI, 2.9–not reached), versus 28.0 months (95% CI, 20.3–39.2) for patients without PD (*p* < 0.001).

There was a significant association between TP53 status and PD at 1 month (*p* = 0.042). Of the 5 patients with PD, 4/5 (80%) had TP53-mutant tumors. No significant association was found between TP53 status and a CR (*p* = 0.628) or PR (*p* = 0.809) at 1 month.

Within the subset of patients with TP53-mutant tumors, imaging follow-up at 1 month demonstrated an ORR of 22/26 (84.6%), inclusive of 11/26 (42.3%) patients with a CR and 11/26 (42.3%) with a PR. Patients with TP53-mutant tumors who exhibited a CR at 1 month had a median OS of 20.3 months (95% CI, 10.0–not reached), versus 15.2 months (95% CI, 9.1–29.3) for patients without a CR, a difference that was not statistically significant (*p* = 0.11). (Figure 3B). Patients with TP53-mutant tumors who exhibited PD at 1 month had a significantly shorter median OS of 3.2 months (95% CI, 2.5–not reached), versus 19.8 months (95% CI, 11.2–52.8) for patients without PD (*p* < 0.001).

Univariate Cox regression analysis of the cohort demonstrated multiple prognostic factors associated with improved OS (Table 4). A TP53 wild-type tumor (*p* = 0.025), AFP concentration ≤ 200 ng/mL (*p* < 0.001), ECOG performance status of 0 (not 1 or 2) (*p* = 0.008), BCLC Stage A or B (not C) (*p* = 0.004), solitary tumor (*p* = 0.013), and CR on initial follow-up imaging (*p* = 0.015) were associated with an improved OS. In subsequent multivariate Cox regression analysis, only an AFP concentration ≤ 200 ng/mL demonstrated statistical significance (*p* = 0.006) (Appendix A).

Competing risk analysis demonstrated that TP53-mutant tumors exhibited a shorter time to local hepatic progression after TAE. The cumulative incidences of local progression were 65.4% (95% CI, 46.0–84.8%) at 6 months and 84.6% (95% CI, 69.4–99.8%) at 12 months for TP53-mutant tumors, versus 40.8% (95% CI, 26.6–55.1%) at 6 months and 55.1% (95% CI, 40.6–69.6%) at 12 months for TP53 wild-type tumors (*p* = 0.0072) (Figure 4A). Neither the time to distant hepatic progression nor the time to extrahepatic progression after TAE demonstrated a significant difference associated with the TP53 status (Figure 4B,D). With regards to any hepatic progression, either local or distant, competing risk analysis demonstrated a shorter time to progression after TAE. The cumulative incidences of any hepatic progression were 73.1% (95% CI, 54.9–91.2%) at 6 months and 92.3% (95% CI, 80.3–100%) at 12 months for TP53-mutant tumors, versus 51.0% (95% CI, 36.5–65.5%) at 6 months and 69.4% (95% CI, 55.9–82.9%) at 12 months for TP53 wild-type tumors (*p* = 0.0083) (Figure 4C).

Grade 2 (moderate) or grade 3 (severe) adverse events related to TAE occurred in seven patients. Five patients had transient acute hepatic dysfunction with ascites and/or encephalopathy. One patient with a history of prior biliary sphincterotomy developed a hepatic abscess that resolved after drainage and antibiotics. One patient had acute pancreatitis managed conservatively and was discharged after a 9-day admission. No grade 4 (life-threatening or disabling) or grade 5 (patient death) adverse events occurred.

## 4. Discussion

A mutation in TP53 was associated with worse OS in patients with HCC treated with TAE, according to the Kaplan–Meier and univariate Cox regression analyses. There is existing evidence that the presence of a TP53 mutation may predict a worse prognosis in HCC, as it is associated with worse survival, a greater likelihood of disease recurrence, and more advanced stages of disease. However, much of the data is derived from patients with early-stage HCC who underwent curative-intent therapies such as resection or transplantation [19,20,21,22,23,29]. For instance, a recent study that included 410 HCC patients treated with liver resection reported that a TP53 mutation is significantly associated with a greater risk of death (*p* = 0.0349) after resection [20]. However, that study only included a small number of patients who received non-curative transarterial locoregional therapy (*n* = 15 treated with TACE).

An AFP concentration (ng/mL) ≤200 was the only statistically significant predictor of an improved OS in our multivariate Cox regression analysis. However, it is important to note that TP53 has a role in the repression of AFP gene expression in HCC, and mutation of TP53 in HCC is associated with elevated AFP [30,31]. The AFP concentration and TP53 status in HCC are thus related covariates that can both be statistically significant in a univariate Cox regression analysis but not in a multivariate analysis. Patients with TP53-mutant HCC were significantly more likely to exhibit PD in initial follow-up imaging than patients with wild type HCC, and PD is associated with a worse OS. Additionally, though a CR in the initial follow-up imaging after TAE was a predictor of a superior OS with respect to the whole cohort, for the subset of patients with TP53-mutant tumors, a CR was not associated with a significantly improved OS. This further supports the association of a TP53 mutation with a worse OS in HCC patients treated with TAE.

A mutation in TP53 was associated with a shorter time to local progression in HCC treated with TAE. A similar result was recently reported by a smaller retrospective analysis of 38 East Asian patients with HBV-related advanced HCC who were treated with TACE and had tumoral tissue analyzed by whole-exome sequencing [32]. TP53-mutant tumors were present in 22/38 (57.9%) patients, and the presence of a TP53 mutation was a significant predictor of TACE failure or refractoriness (*p* = 0.020). The study did not report survival outcomes. In contrast to that study, the Western cohort of the current study did not have HBV as the predominant etiology of liver disease. An association between TP53 mutation and a greater risk of recurrence after resection (*p* = 0.028) of HCC has also been reported [20]. A mutation in TP53 may predict a greater risk of progression or recurrence regardless of the treatment modality.

The patients in the cohort with TP53-mutant tumors were more likely to have macrovascular invasion and more advanced disease in terms of the BCLC stage than TP53 wild-type patients. It has been reported that a TP53 mutation is more prevalent in advanced-stage HCC, and it is hypothesized that the mutation may promote disease progression [19,20]. Within a predominantly surgical cohort, TP53 mutation was present in 35% of BCLC C tumors versus 15.5% to 17.3% of BCLC 0 to BCLC B tumors (*p* < 0.0001) [20].

Efforts to stratify HCC by making genotype–phenotype correlations with transcriptomic analysis support a link between a TP53 mutation, aggressive disease, and worse prognosis. A recent study integrated 16 previously established transcriptomic subtypes of HCC and identified 5 distinct consensus subtypes, A–E [33]. Subtype A had the greatest prevalence of TP53 mutations and was associated with the worst prognosis and highest rate of microvascular invasion. A separate transcriptomic analysis of HCC identified six subtypes, G1–G6 [29]. Both subtypes G2 and G3 had a high frequency of TP53 mutations and were associated with a poor prognosis, with G3 having the poorest prognosis. Subtype G2 was associated with HBV infection and G3 was not. While the current study lacked transcriptomic analysis, due to the high frequency of TP53 mutations within the Western cohort where HBV was not the predominant etiology of liver disease, it is suspected that subtype G3 was highly prevalent. The link with transcriptomic subtypes also suggests that the etiology of HCC may be an associated prognostic factor.

TP53 is a tumor suppressor and multifunctional transcription factor that helps mediate cellular responses to hypoxia [34]. The precise interplay between TP53 and the response of HCC to acute hypoxia in the setting of ischemia-inducing transarterial therapy, such as TAE or TACE, remains to be determined. However, there is evidence to suggest that TP53-mutant tumors may be resistant to ischemia-based treatments. A mouse model study utilizing human colon adenocarcinoma cell lines differing only in TP53 status found that a loss or inactivation of TP53 promotes hypoxia-induced angiogenesis, in part through increased levels of hypoxia-inducible factor 1α (HIF-1α) [35]. It is known that ischemia of HCC leads to activation of hypoxia-inducible factors, and greater expression of HIF-1α has been associated with a worse OS and greater likelihood of recurrence in HCC patients treated with surgical resection [36,37]. The resistance to ischemia-based treatments for HCC conferred by a TP53 mutation may account for the worse survival and earlier recurrence after TAE.

It remains to be determined if there is an association between TP53 status and clinical outcomes in HCC treated with yttrium-90 (Y90) transarterial radioembolization (TARE), which does not rely on ischemia as the primary mechanism of tumor cell death. It is possible that treatment of patients with TP53-mutant HCC using TARE may yield better outcomes than TAE. Given the propensity for local progression, the initiation of systemic therapy in TP53-mutant HCC patients, regardless of the initial response to TAE, may be of benefit. Additionally, there are investigational approaches to restoring TP53 function, including small-molecule reactivators of TP53 and adenoviral transfection, which offer the potential to improve outcomes in patients with TP53-mutant HCC.

The increasing availability and utilization of tumor molecular profiling has led to the identification of prognostic molecular markers in cancers, some of which are clinically actionable. In colorectal cancer, a KRAS mutation portends worse survival and a shorter time to recurrence and influences the treatment strategy [38]. In intrahepatic cholangiocarcinoma, the second most common primary hepatic malignancy, the presence of a high-risk gene signature, which includes alterations in TP53, KRAS, or CDK2NA, has been associated with a worse OS in patients treated with medicine, surgery, or TARE [39,40]. While there are currently no molecular markers routinely used in clinical practice to guide the management of patients with HCC, TP53 is a candidate that should be further investigated.

This study is primarily limited by its sample size and partially retrospective nature. The high specificity and positive predictive value of dynamic computed tomography or magnetic resonance imaging tailored to the diagnosis of HCC in cirrhotic patients obviates the clinical necessity of a biopsy in many cases [41,42]. As such, the availability of tissue for a genetic analysis in HCC patients treated without surgery is limited.

## 5. Conclusions

A mutation in TP53 may predict a worse OS and shorter time to local progression in HCC patients treated with TAE. Further investigation is needed to validate these findings and determine the optimal treatment strategy for patients with TP53-mutant HCC.

## Figures and Tables

**Figure 1 curroncol-32-00051-f001:**
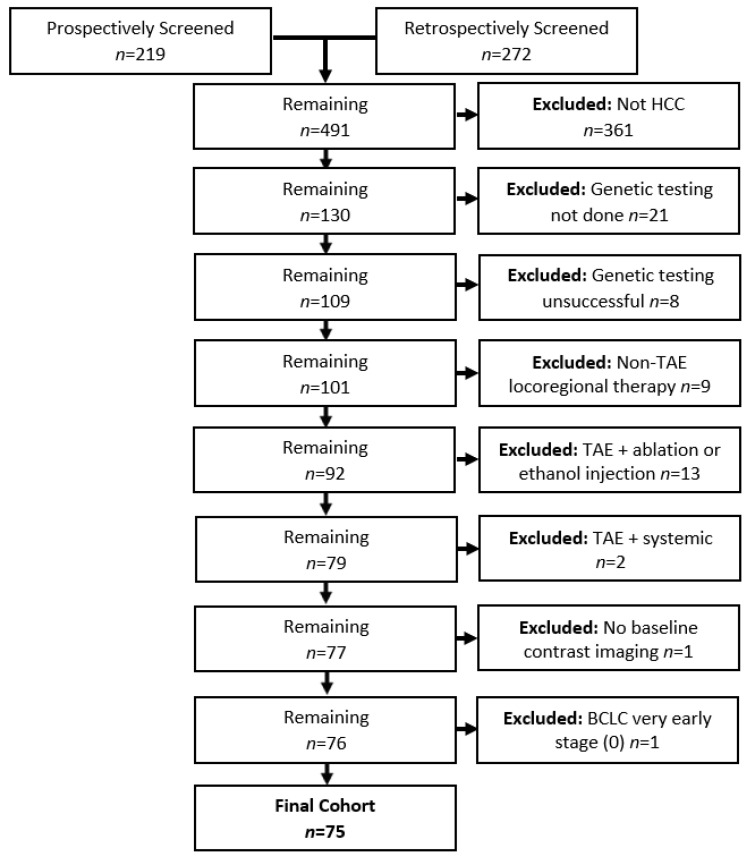
Flowchart of screening the study cohort.

**Figure 2 curroncol-32-00051-f002:**
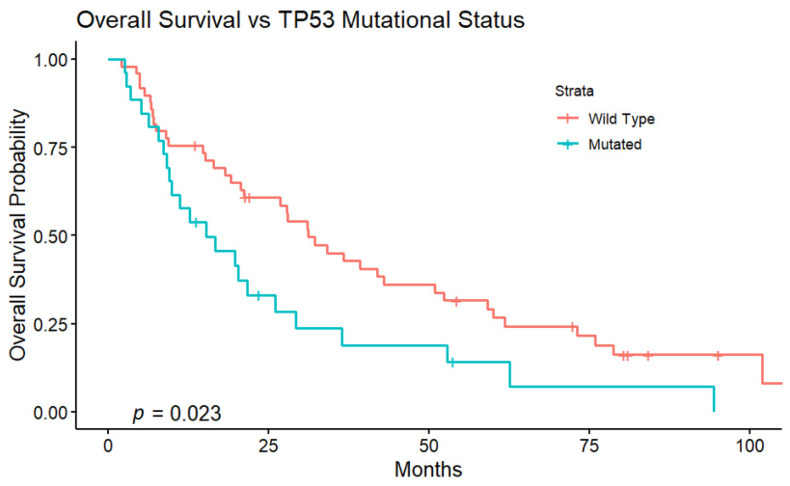
Patients with TP53-mutant HCC had significantly worse OS.

**Figure 3 curroncol-32-00051-f003:**
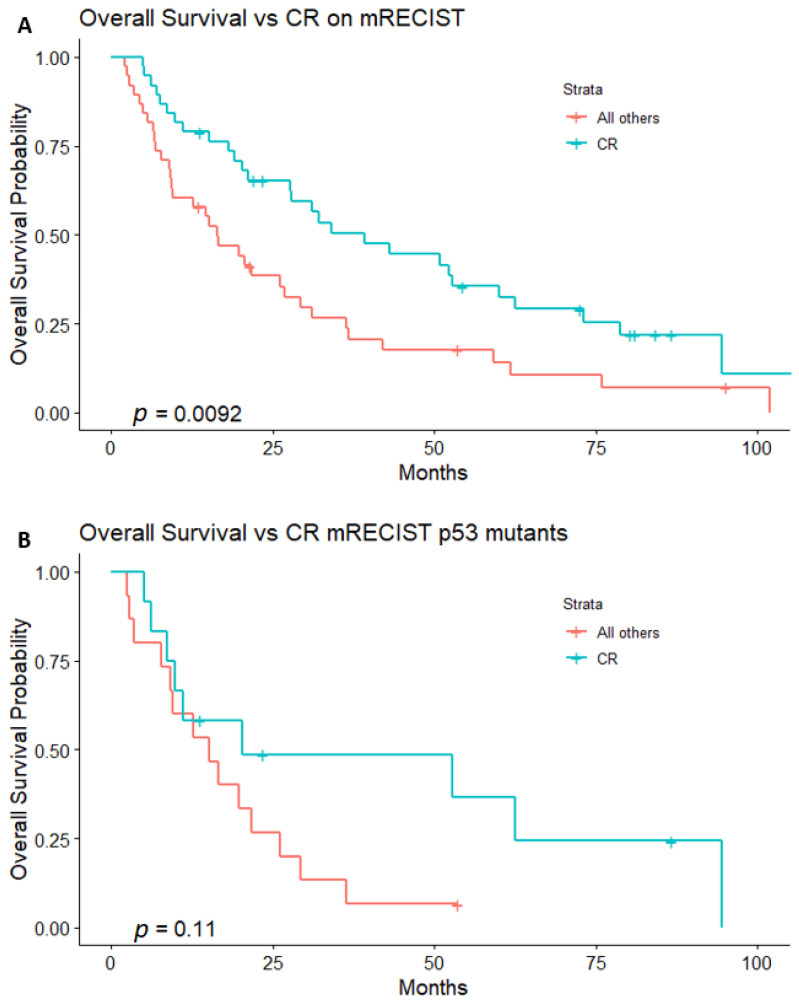
(**A**). For the entire cohort, patients who exhibited a CR had a significantly longer OS. (**B**). For the subset of patients with TP53-mutant tumors, a CR was not associated with a significantly longer OS.

**Figure 4 curroncol-32-00051-f004:**
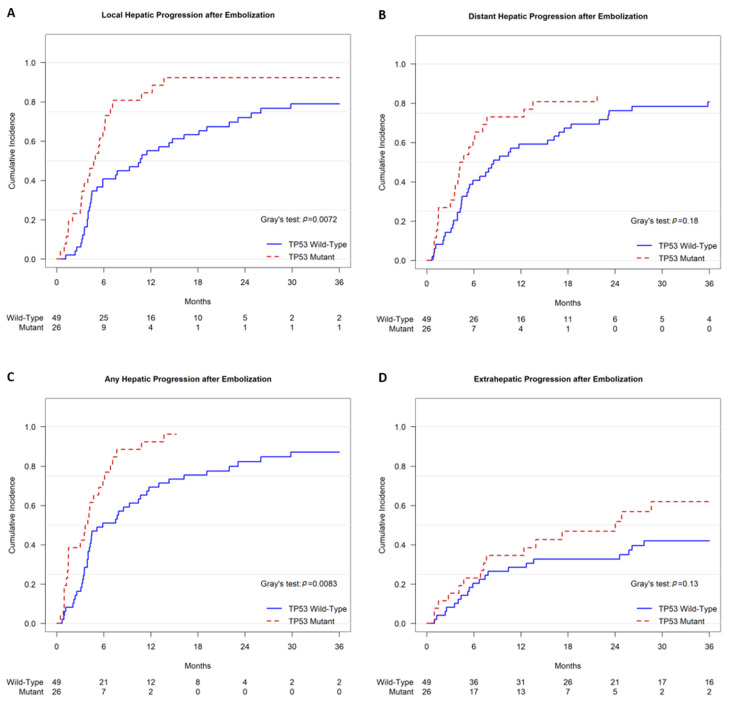
Cumulative incidence functions of progression after TAE. Patients with TP53-mutant HCC had significantly shorter times to local hepatic progression (**A**) and any hepatic progression (**C**) after TAE. There was no significant association between TP53 status and time to distant hepatic progression (**B**) or extrahepatic progression (**D**).

**Table 1 curroncol-32-00051-t001:** Demographics and clinical characteristics.

Characteristic	All Patients (*n* = 75)	TP53 Mutant (*n* = 26)	TP53 Wild-Type (*n* = 49)	*p*-Value
Age, Median (IQR), Years	70.0 (62.0–76.3)	69.4 (64.5–74.5)	70.0 (60.7–76.9)	0.925
Male Gender, No (%)	63	23 (88.5)	40 (81.6)	0.526
Ethnicity, No (%)				0.340
White	57	19 (73.1)	38 (77.6)	
Asian	7	2 (7.7)	5 (10.2)	
Black	5	1 (3.8)	4 (8.2)	
Other	2	2 (7.7)	0	
Unknown	4	2 (7.7)	2 (4.1)	
Other Primary Cancer		8 (30.8)	8 (16.3)	0.134
Etiology, No (%)				0.982
Hepatitis B	6	3 (11.5)	3 (6.1)	
Hepatitis C	27	8 (30.8)	19 (38.8)	
Steatohepatitis	12	4 (15.4)	8 (16.3)	
Alcohol	6	3 (11.5)	3 (6.1)	
Other	4	1 (3.8)	3 (6.1)	
Multiple	7	3 (11.5)	4 (8.2)	
Unknown	13	4 (15.4)	9 (18.4)	
ECOG PS, No (%)				0.060
0	54	15 (57.7)	39 (79.6)	
1 or 2	21	11 (42.3)	10 (20.4)	
Prior HCC Treatment, No (%)	17	3 (11.5)	14 (28.6)	0.339
Surgery	12	2 (7.7)	10 (20.4)	
Image-guided Locoregional	6	1 (3.8)	5 (10.2)	
Systemic	2	1 (3.8)	1 (2)	
Child–Pugh Class, No (%)				0.658
A (Score 5–6)	69	25 (96.1)	44 (89.8)	
B (Score 7–11)	6	1 (3.8)	5 (10.2)	
ALBI Grade, No (%)				0.757
Grade 1 (Score ≤ −2.60)	39	15 (57.7)	24 (49)	
Grade 2 (−2.60 < Score ≤ −1.39)	35	11 (42.3)	24 (49)	
Grade 3 (−1.39 < Score)	1	0	1 (2)	
BCLC Stage, No (%)				0.007
A or B	40	8 (30.8)	32 (65.3)	
C	35	18 (69.2)	17 (34.7)	
AFP, Median (IQR), ng/mL	16.5 (6–296.6)	18.4 (5.7–685.2)	15.3 (7.1–166.3)	0.297
Histologic Grade, No (%) *				0.011
Well or Moderately Differentiated	59 (78.7)	16 (61.5)	43 (87.8)	
Poorly Differentiated	6 (8)	5 (19.2)	1 (2)	

* Histologic grade of 10 tumors was not mentioned within the pathology report.

**Table 2 curroncol-32-00051-t002:** Baseline tumor characteristics.

Characteristic	All Patients (*n* = 75)	TP53 Mutant (*n* = 26)	TP53 Wild-Type (*n* = 49)	*p*-Value
Lesion Number, No (%)				0.811
Single	18	5 (19.2)	13 (26.5)	
≤3	40	13 (50)	27 (55.1)	
>3	35	13 (50)	22 (44.9)	
Bilobar Disease	38	15 (57.7)	23 (46.9)	0.472
Lesion Diameter, Median (IQR), cm	5.8 (3.9–8.7)	7.9 (4.2–9.3)	5.8 (3.6–6.4)	0.261
Macrovascular Invasion, No (%)	15	9 (34.6)	6 (12.2)	0.018
Extrahepatic Disease, No (%)	5	3 (11.5)	2 (4.1)	0.334

**Table 3 curroncol-32-00051-t003:** Imaging response according to mRECIST at 1 month post-TAE.

Imaging Response at 1 Month Follow-Up	All Patients (*n* = 75)	TP53 Mutant (*n* = 26)	TP53 Wild-Type (*n* = 49)	*p*-Value
Complete Response	36	11	25	0.628
Partial Response	34	11	23	0.809
Stable Disease	0	0	0	-
Progressive Disease	5	4	1	0.042

**Table 4 curroncol-32-00051-t004:** Prognostic factors of improved overall survival, according to the univariate analysis.

	Univariate Cox Analysis		
	HR	95% CI	*p*-Value
TP53 Status (Wild-Type vs. Mutant)	0.55	0.32–0.93	0.025
AFP (ng/mL) (≤200 vs. >200)	0.32	0.18–0.56	<0.001
ECOG PS (0 vs. 1 or 2)	0.46	0.26–0.82	0.008
BCLC Stage (A or B vs. C)	0.47	0.28–0.81	0.004
Lesion Size (≤3 cm vs. >3 cm)	0.50	0.24–1.01	0.053
ALBI Grade (1 vs. >1)	0.94	0.57–1.56	0.815
Child–Pugh Class (A vs. B)	0.62	0.26–1.45	0.266
Tumor Number			
(1 vs. >1)	0.43	0.22–0.84	0.013
(≤3 vs. >3)	0.65	0.39–1.08	0.093
Tumor Distribution (Unilobar vs. Bilobar)	0.61	0.37–1.02	0.057
Baseline Macrovascular Invasion (No vs. Yes)	0.61	0.34–1.12	0.111
Histologic Grade (Well or Moderately Differentiated vs. Poorly Differentiated)	0.76	0.32–1.79	0.529
Initial mRECIST Response (CR vs. not CR)	0.54	0.33–0.9	0.015

## Data Availability

The datasets presented in this article are not readily available because the prospectively collected data are part of an ongoing study. Requests to access the datasets should be directed to zhaok@mskcc.org.

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
