# Peer review of "TP53 Mutation Predicts Worse Survival and Earlier Local Progression in Patients with Hepatocellular Carcinoma Treated with Transarterial Embolization"

_curroncol, 2025, doi:10.3390/curroncol32010051_

Round 1

Reviewer 1 Report

Comments and Suggestions for Authors

Interesting paper with a relevant topic, investigating TP53 as a biomarker predictive of OS in HCC patients treated with TAE.

ethics approval obtained. 

Weeknesses:

- The inclusion of more datasets, especially those from different hospitals, could enhance the study's generalizability. particularly, other treatments than TAE could have been included. why did you only include TAE and not other locoregional therapies?

- Exploring additional imputation methods could add depth to the analysis. specifically, the authors performed only a univariate analysis but no multivariate analysis. in the UVA, multiple factors (most of them known to be prognostically relevant) including TP53 were identified as predictors of OS. however, a multivariate analysis needs to be performed to see if TP53 remains relevant in this question; parituclalry, given the heterogenous cohort that also included extra hepatic disease, microvascular invasion, impaired performance status, and BCLC C, and impaired liver function, all of which are known to be associated with poor prognosis. 

- define years of experience of radiologists who did the interventions and the readings / response assessments. 

- Expanding on how the findings could be applied in real-world scenarios might improve the practical relevance of the research. please discuss in more details

- the use of TAE instead of TACE likely reflects both institutional preferences and the specific goals of the study, which was focused on the effects of embolization and genetic markers rather than the addition of chemotherapy agents used in TACE. The center is known for their performance of TAE. however, as the majority of centers worldwide prefers TACE over TACE, please add a more in-depth-discussion of the findings expected after TACE compared to TAE, and how this lack of knowledge in your study is a limitation. 

Reviewer 2 Report

Comments and Suggestions for Authors

This study aimed to evaluate the associations between TP53 status and outcomes following transarterial embolization for hepatocellular carcinoma treatment. Patients who underwent TAE and genomic analysis of tumoral tissue were studied. The primary outcome was overall survival and correlation to TP53 status, and the secondary outcome was time to progression. The association with TP53 status was evaluated using the Gray test. Overall, 75 patients were included of whom 26 (34.7%) had TP53 mutant HCC. They found that patients with TP53 mutant HCC had significantly worse median OS of 15.2 months versus 31.2 months for TP53 wild-type HCC. Competing risk analysis showed a shorter time to local hepatic progression (at the site of previously treated tumor) after TAE in patients with TP53 mutant HCC. Cumulative incidences of local progression at 6 and 12 months for TP53 mutant HCC were 65% and 84% vs 40 % and 55% for TP53 wild-type HCC (p=0.007). They therefore concluded that a TP53 mutation may predict worse OS and shorter time to local progression in HCC patients treated with TAE.

The study is interesting since it would provide a prognostic marker to predict treatment outcomes and prognosis. However, some information is needed to support the study's conclusions.

-Since this is a single-institution study, and patients underwent biopsy at the time of TAE which is not a standard of care, ethical committee approval should be declared and reported.

-Study population: according to Table 1 data, the majority of patients (n=27) had underlying Hepatitis C-related liver disease, and 30.8% of them exhibited TP53 mutant, while there were 13 patients with unknown etiology of underlying liver disease, 15.4% of whom had TP53 mutant. It seems that HCV patients might have a significantly higher rate of TP53 mutant compared with other liver disease etiologies. This might suggest an etiology-related prognostic factor.

-TAE procedure: could the authors describe whether they used a selective/superselective transarterial approach? Were there treatment-related adverse events?

-Table 3 (Imaging response per mRECIST at 1 month post-TAE): in a study assessing the prognostic significance of a genetic factor, it would be of clinical relevance to describe the prognostic significance of clinical/laboratory parameters according to recent literature data. In particular, it has been recently reported that a post transarterial treatment-related hypertransaminasemia, is significantly correlated with the objective radiological response thus offering a simple tool to predict treatment response and for a tailored treatment approach, as recently demonstrated (J Pers Med. 2021 Oct 17;11(10):1041. doi: 10.3390/jpm11101041). Could the authors report such information?

is a si

ngle-institution study.

Reviewer 3 Report

Comments and Suggestions for Authors

The authors enrolled 75 HCC patients and performed genetic analysis for TP53 mutation, and found that TP53 mutation may predict worse OS and shorter time to local progression in HCC patients treated with TAE. There are several comments:

1.    Table 1 and 2 showed clinical characteristics of TP53 mutant patients and TP53 wild-type patients. Patients with TP53 mutation had more advanced HCC stage, poorer histologic grade and more macrovascular invasion. All these factors influence the prognosis of HCC patients and may cause bias in results. The author may use propensity score matching analysis to reduce the bias.

2.    Table 4 presented the results of univariate Cox regression analysis. However, macrovascular invasion and histologic grade had no influence on OS (p>0.05) in this study, which was very confusing and contrary to common sense. How to explain it? Why did the authors do not perform multivariate Cox regression analysis?

3.    Is there any external validation cohort to testify the conclusions of this study, such as some published research databases?

4.    Most patients had multiple HCC in this study. Did the authors perform genetic analysis for each lesion?

5.    There is a lack of description for Figure 3B in results. The title of the figures may cause confusion for readers.

6.    It is clearly common knowledge that patients with PD have shorter OS than patients without PD. The results in Figure 4 are not necessary to present and have no effect on supporting the conclusion of this study.

Round 2

Reviewer 3 Report

Comments and Suggestions for Authors

The authors did not addressed my concerns in their revision.